# Environmental Toxin Biliatresone-Induced Biliary Atresia-like Abnormal Cilia and Bile Duct Cell Development of Human Liver Organoids

**DOI:** 10.3390/toxins16030144

**Published:** 2024-03-11

**Authors:** Yue Hai-Bing, Menon Sudheer Sivasankaran, Babu Rosana Ottakandathil, Wu Zhong-Luan, So Man-Ting, Chung (Patrick) Ho-Yu, Wong (Kenneth) Kak-Yuen, Tam (Paul) Kwong-Hang, Lui (Vincent) Chi-Hang

**Affiliations:** 1Department of Surgery, School of Clinical Medicine, The University of Hong Kong, Hong Kong SAR, China; yuehaibingchina@126.com (Y.H.-B.); drsudhimenon@omail.edu.pl (M.S.S.); rosana.ottakandathilbabu@ndm.ox.ac.uk (B.R.O.); hannawu@hku.hk (W.Z.-L.); jayso@hku.hk (S.M.-T.); chungphy@hku.hk (C.H.-Y.); kkywong@hku.hk (W.K.-Y.); pkhtam@must.edu.mo (T.K.-H.); 2Dr. Li Dak-Sum Research Centre, The University of Hong Kong, Hong Kong SAR, China; 3Faculty of Medicine, Macau University of Science and Technology, Macau SAR 999078, China

**Keywords:** biliatresone, biliary atresia, organoids

## Abstract

Biliary atresia (BA) is a poorly understood and devastating obstructive bile duct disease of newborns. Biliatresone, a plant toxin, causes BA-like syndrome in some animals, but its relevance in humans is unknown. To validate the hypothesis that biliatresone exposure is a plausible BA disease mechanism in humans, we treated normal human liver organoids with biliatresone and addressed its adverse effects on organoid development, functions and cellular organization. The control organoids (without biliatresone) were well expanded and much bigger than biliatresone-treated organoids. Expression of the cholangiocyte marker CK19 was reduced, while the hepatocyte marker HFN4A was significantly elevated in biliatresone-treated organoids. ZO-1 (a tight junction marker) immunoreactivity was localized at the apical intercellular junctions in control organoids, while it was markedly reduced in biliatresone-treated organoids. Cytoskeleton F-actin was localized at the apical surface of the control organoids, but it was ectopically expressed at the apical and basal sides in biliatresone-treated organoids. Cholangiocytes of control organoids possess primary cilia and elicit cilia mechanosensory function. The number of ciliated cholangiocytes was reduced, and cilia mechanosensory function was hampered in biliatresone-treated organoids. In conclusion, biliatresone induces morphological and developmental changes in human liver organoids resembling those of our previously reported BA organoids, suggesting that environmental toxins could contribute to BA pathogenesis.

## 1. Introduction

Biliary atresia (BA) is a devastating inflammatory cholangiopathy characterized by biliary cirrhosis, fibrotic disorder, neonatal cholestasis and inflammatory obstruction of the biliary tract [1,2]. Kasai surgery is the first-line treatment for BA patients, which replaces the obliterated extrahepatic bile duct with an intestinal conduit to re-establish bile flow (for review, see [3]). However, for many BA patients, liver transplantation is the final outcome [4]. BA affects 5–20:100,000 live births. Over 90% of the cases are non-familial and without a clear genetic cause (for review see [5]). Susceptibility loci have been identified on chromosomes 10q24.2 [6] and 2q37.3 [7]. Later, the *ADD3* and *EFEMP1* genes were identified as aberrantly regulated in BA [8,9]. Gene mutations in the Polycystic kidney disease 1 like 1 Gene (PKD1L1) were also identified in patients with BA splenic malformation syndrome [10]. BA has multiple etiologies [11], and viral infections [12], exposure to toxins [13] and a dysfunctional immune response [14] are likely to contribute by causing inflammation and driving BA pathogenesis.

There are multiple known general biliary toxins, as well as natural toxic causes of organ fibrosis, and thus, the role of a biliary toxin in BA would not be surprising [15]. The toxin biliatresone, a naturally occurring isoflavonoid-related 1,2-diaryl-2-propenone found in *Dysphania glomulifera* and *D. littoralis*, has been implicated in BA-like syndrome outbreaks in Australian livestock [13,16]. It also causes bile duct destruction in zebrafish [13], mouse cholangiocyte spheroids [17,18], ex vivo bile duct culture [18] and mouse neonates [19,20]. Biliatresone contains a reactive α-methylene ketone group, is highly electrophilic, and binds to reduced glutathione (GSH) [21,22,23]. GSH is the most abundant endogenous small molecule antioxidant, and decreased levels of GSH have been shown to be a key part of the mechanism of action of biliatresone [18,24]. However, evidence of an adverse effect of biliatresone on the growth of human cholangiocytes and of an etiological/pathobiological role of biliatresone exposure in BA in humans is still missing.

Cells from the cholangiocyte lineage of mice and humans can be cultured in the form of 3-D organoids [25,26,27,28]. The liver organoids are composed of cholangiocytes and hepatocytes, with cholangiocytes being the major cell type in the organoids. Liver organoids from BA liver and RRA-BA (Rhesus rotavirus A-infected mouse) mouse liver exhibited (a) aberrant morphology with disturbed apical-basal organization, (b) a shift from cholangiocyte to hepatocyte transcriptional signatures and (c) altered beta-amyloid-related gene expression [27]. The aberrant organoid morphology appears specific for BA, indicating that human liver organoids are a good human proxy for patho-mechanistic studies of BA.

BA is a rare disease of the liver and bile ducts that occurs in infants. We generated liver organoids from liver biopsies of age-matched non-BA infants to (i) address whether biliatresone has an adverse effect on the differentiation and functions of human cholangiocytes and (ii) provide evidence for an etiological/pathobiological role of biliatresone exposure in BA in humans.

## 2. Results

### 2.1. Biliatresone-Induced Aberrant Growth of Human Liver Organoids

To test if biliatresone influenced the growth of human liver organoids, we added 2 µg/mL biliatresone to organoid cultures and studied the growth rate, morphology and histology of organoids grown with or without biliatresone (Figure 1). Control organoids expanded and developed into a well-expanded spherical shape with a single-cell layer of epithelial cells and a single vacuole inside (Ctrl; with DMSO) from Day 0 to Day 5 (Figure 1B). In contrast, there were fewer organoids in the day 5 culture with biliatresone. The organoids were generally smaller and not well-expanded (arrows) or very tiny and poorly expanded and had thick cell layers with multiple vacuoles (arrowheads; Figure 1B). To study the growth rate of control and biliatresone-treated organoids, the diameters of organoids in control and biliatresone treatment were determined on Days 0, 3 and 5 of culture (Figure 1C). Without the addition of biliatresone, the diameter of liver organoids increased from 300 ± 70 µm (Mean ± SD) on Day 0 to 680 ± 190 µm (Mean ± SD) on Day 5 of incubation. In contrast, in the culture with 2 µg/mL biliatresone, the diameter of liver organoids decreased from 350 ± 28 µm (Mean ± SD) on Day 0 to 215 ± 74 µm (Mean ± SD) on Day 5 of incubation. The sizes of bilitresone-treated organoids on Days 3 and 5 were statistically significantly smaller (*p* < 0.05) than those of control organoids on the respective days.

### 2.2. Biliatresone-Induced Hepatocytic Differentiation of Human Liver Organoids

Transcriptomic analysis of BA organoids has revealed a shift from cholangiocyte to hepatocyte transcriptional signatures [27,29], and biliatresone has been shown to induce BA-like phenotypes in mice [20] and zebrafish [13]. We investigated if biliatresone perturbed the differentiation of cholangiocytes and hepatocytes in human liver organoids. Hepatocytic differentiation in human organoids was assessed via immuno-flurosecence for hepatocyte makers (HNF4A). Expression of HFN4A was only detected in very few cells in the control organoids; in contrast, HNF4A expression was markedly elevated in many cells in biliatresone-treated organoids (Figure 2A). The percentage of HNF4A+ cells in organoids increased statistically significantly (*p* < 0.05) from 14.9 ± 2% (Mean ± SD) in control organoids to 57.3 ± 8% (Mean ± SD) in biliatresone-treated organoids. Furthermore, the expression of the cholangiocyte marker MDR1 was drastically reduced in biliatresone-treated organoids (Figure 2B). To further confirm the defective cholangiocyte differentiation of biliatresone-treated organoids, we performed an MDR1-mediated Rhodamine 123 (R123) transport assay. In control organoids, no fluorescence signal was detected outside the organoids, but a very strong fluorescence signal (around 50,000 units) was detected inside the organoids at 30 min of incubation (Figure 2C), indicating that control organoids were efficient in transporting R123 from the medium into the organoids. In contrast, in biliatresone-treated organoids, the fluorescence signal intensities outside and inside the organoids were similar (between 10,000 and 20,000 units) at 30 min of incubation (Figure 2C), indicating that biliatresone-treated organoids were inefficient in transporting R123 across the organoid membrane, and only a low level of R123 was accumulated within the organoids after incubation. All the above indicated that biliatresone could induce the defective cholangiocyte differentiation of human liver organoids.

### 2.3. Biliatresone-Induced Apical-Basal Polarity Defect in Human Liver Organoids

Disturbed apical-basal and cytoskeleton organization have also been observed in BA liver organoids [27] as well as in biliatresone-treated mouse cholangiocytes [18]. To test if biliatresone induced apical-basal polarity defects in human organoids, we stained the control and biliatresone-treated organoids with an anti-F-actin antibody and anti-ZO-1 antibody. F-actin immunoreactivity was localized at the apical side of the control organoids (Figure 3A), but F-actin was now localized to both the apical and the basal sides of the biliatresone-treated organoids. In line with the perturbed cytoskeleton in biliatresone-treated organoids, ZO-1 immuno-reactivity was markedly reduced and without a clear apical predominance in biliatresone-treated organoids (Appendix A), which is in contrast to the predominance ZO-1 staining at the apical side of the control organoids (Appendix A). Zona occludens 1 (ZO-1) stabilizes the tight junction solute barrier through coupling to the perijunctional cytoskeleton, and the depletion of ZO-1 leads to defects in the barrier for large solutes [30]. To test if biliatresone-treated organoids have defects in the tight junction barrier for solutes, we incubated control and biliatresone-treated organoids with FITC-dextran. FITC-dextran is a fluorescently labeled small molecule commonly used for measuring the permeability of cell monolayers. In control organoids, the fluorescence signal intensity outside the organoids was between 30,000 and 40,000 units, whereas it was much lower (around 18,000 units) inside the organoids at 10 min of incubation (red; Figure 3B). Furthermore, the fluorescence signal intensity gradient along the line of the measurement at 30 min of incubation (green) was identical to that at 10 min of incubation (red), indicating that no further FITC-dextran molecules entered into the organoids during the additional 20 min of incubation (Figure 3B). In contrast, in biliatresone-treated organoids, the fluorescence signal intensities outside and inside the organoids were similar (between 20,000 and 30,000) at 10 min of incubation (red; Figure 3B). The fluorescence signal intensity inside the biliatresone-treated organoids at 30 min of incubation (green) was higher than that at 10 min of incubation (red), indicating that more FITC-dextran molecules entered into the organoids during the additional 20 min of incubation (Figure 3B). All the above results indicated that the single epithelial cell layer of the control organoids provided a structural barrier preventing the diffusion of small molecules such as FITC-dextran into the organoids, and the structural barrier of the organoids was compromised by the biliatresone treatment.

### 2.4. Biliatresone Decreased the Number of Ciliated Cholangiocytes in Human Organoids

Shorter, misoriented, or less abundant cholangiocyte cilia were commonly observed in several studies of both syndromic and non-syndromic BA patients [31,32,33], suggesting that cilia defects could contribute to BA. To investigate if biliatresone disturbed ciliogenesis in human organoids, we immunostained control and biliatresone-treated organoids for ciliary proteins Pericentrin (PCNT) and alpha-tubulin. As shown in Figure 4, the percentages of ciliated cells increased gradually in control organoids from 10.7 ± 5% (Mean ± SD) on Day 1 to 56.4 ± 10% (Mean ± SD) on Day 5. However, in biliatresone-treated organoids, the percentages of ciliated cells were only 4.2 ± 2% (Mean ± SD) on Day 1 and dropped to 0% on Day 3, which was statistically significantly lower (*p* < 0.05) than that of control organoids (Figure 4).

Cilia acts as a mechanical sensor for fluid flow in the bile duct; shear stress could induce an influx of calcium in cholangiocytes. To functionally assay cilia sensory function in cholangiocytes, we dissociated organoids and seeded the cells onto a chip, allowing them to grow into a monolayer. ZO-1 staining confirmed the formation of a cholangiocyte monolayer with their apical sides facing toward the medium (Appendix A). Cholangiocytes remained quiescent under static conditions and showed increased calcium influx in response to medium flow (Appendix A). Thus, we used the cholangiocyte monolayer to address if cilia mechanosensory function was disrupted after biliatresone treatment. As shown in Figure 5A,B, medium flow induced a calcium influx and an increase in fluorescence intensity in control cholangiocytes. In contrast, biliatresone-treated cholangiocytes failed to respond to medium flow and remained quiescent. The percentage of evoked cells after perfusion stimulation of control (Ctrl) cholangiocytes was statistically significantly higher (*p* < 0.05) than that of biliatresone-treated (BTS) cholangiocytes (Figure 5C). There was approximately 28 ± 8% (Mean ± SD) of evoked cholangiocytes after perfusion in the control culture, while only 1.3 ± 1% (Mean ± SD) of evoked cholangiocytes in the biliatresone culture after perfusion. The green fluorescence microscope video recording of control and biliatresone-treated cholnagiocytes after perfusion is shown in Appendix A. 

## 3. Discussion

The etiology of BA is still poorly understood. Biliatresone, a plant isoflavonoid-related 1,2-diaryl-2-propenone, has been implicated in BA-like syndrome outbreaks in Australian livestock [13,16]. It also causes bile duct destruction in zebrafish [13], mouse cholangiocyte spheroids [17,18], ex vivo bile duct culture [18], and mouse neonates [19,20]. However, evidence of the adverse effect of biliatresone on the growth of human cholangiocytes is still lacking. This study investigated the effects of the biliatresone toxin on human cholangiocytes. Using human liver organoids, we showed that biliatresone induced dysmorphogenesis, disturbed apical-basal and cytoskeleton organization, defective development, reduction in primary cilia and impaired cilia mechanosensory function of cholangiocytes. Our study provides the first evidence showing that biliatresone induces BA-like morphological and developmental changes in human cholangiocytes, suggesting that environmental toxins could contribute to BA pathogenesis in humans.

Biliatresone treatment induces developmental defects of human liver organoids similar to the defects observed in liver organoids derived from BA liver tissue [27,29]. These defects include retarded growth, disturbed apical-basal organization and defective cholangiocyte development. We observed a reduction in cholangiocyte MDR transport, a reduction in ciliated cholangiocyte and impaired cilia sensory response in biliatrsone-treated organoids, which further corroborates a defective cholangiocyte development owing to biliatresone treatment. In line with our observations, disrupted apical-basal polarity [18] and a reduced number of primary cilia [13] in cholangiocytes were reported in biliatresone-treated mouse cholangiocyte organoids.

Cholangiocytes, like many other cells, present primary cilia extending from their apical plasma membranes into the bile duct lumen. These cilia act as flow-stress sensors to detect changes in bile flow, composition and osmolality and adjust the secretion of bile acid and CO_3_^2−^ accordingly [34,35,36]. Defects in ciliary function could cause developmental or degenerative disorders across a wide spectrum [37]. Mutations in genes encoding cholangiocyte ciliary-associated proteins, such as polycystin-1 and -2 and fibrocystin, cause biliary dysgenesis, hepatic fibrosis and cystogenesis in liver diseases such as Autosomal Dominant Polycystic Kidney Disease (ADPKD) and Autosomal Recessive PKD (ARPKD). Our recent whole-genome sequencing study detected rare, deleterious de novo or biallelic variants of liver-expressed ciliary genes, including *PCNT*, *KIF3B* and *TTC17*, in around 30% of non-syndromic BA patients [38]. Cholangiocyte cilia act as mechano-, chemo-, and osmo-sensors for bile flow and regulation of bile formation in bile ducts. Through the ciliary-specific and associated receptors, ion channels and sensory signaling molecules on the ciliary membrane, changes in bile content are detected, and signals are transduced to regulate cholangiocyte function. It was found that bile acids induce hyper-proliferation of non-ciliated cholangiocytes but not of ciliated cholangiocytes [39]. Besides bile acids, there are neuropeptides, neurotransmitters and hormones that promote cholangiocyte hyper-proliferation during cholestasis, whereas cholangiocyte hyper-proliferation could promote hepatic fibrosis [40]. Shorter, abnormally oriented and less abundant cilia were observed in the intra-hepatic cholangiocytes of syndromic and non-syndromic BA patients [32,33]. In extra-hepatic cholangiocytes, cilia were almost completely absent in BA neonates [31]. Fibrocystin, a protein that localizes to cholangiocyte cilia and the basal body of bile ducts, was reduced in BA livers [41]. All the above findings indicate that abnormal cilia and dysregulated cilia functions of cholangiocytes are intimately associated with the development of BA. Exposure of bile duct cells to toxins such as biliatresone may lead to a reduction in ciliated cholangiocyte and dysregulated sensory responses to bile acids or other regulators, which could then lead to cholangiocyte hyper-proliferation and hepatic fibrosis in BA.

Biliatresone decreased the cellular antioxidant glutathione (GSH) and *SOX17* levels of mouse cholangiocytes [18]. GSH and *SOX17* are essential for the stabilization of the cellular cytoskeleton, which plays a vital role in maintaining the cellular polarity and in ciliogenesis [13,18]. *SOX17* is expressed in the endoderm from the onset of gastrulation and plays essential roles in cholangiocyte and bile duct development [42,43]. *Sox17* heterozygous mice develop a BA-like phenotype as the gallbladder epithelium becomes detached from the luminal wall, which indicates that *Sox17* is required to maintain the epithelial architecture of the gallbladder and cystic duct [42]. A small but significant drop in *SOX17* expression was detected in Day-2 biliatresone-treated organoids (Appendix A), which was associated with perturbations of the cytoskeleton, apical-basal polarity, ciliogenesis defects, and loss of epithelial integrity in human cholangiocytes, as revealed by abnormal F-actin, α-tubulin and ZO-1 immunofluorescence (Figure 3, Figure 4 and Appendix A). We also observed a small but significant drop in GSH levels in biliatresone-treated organoids at 3 h, but the GSH level was restored in biliatresone-treated organoids at 6, 24 and 48 h (Appendix A). In line with our observation of the transient drop in GSH in our biliatresone-treated organoids, biliatresone has also been shown to cause a rapid and transient decrease in GSH, which was both necessary and sufficient to mediate its effects in mouse cholangiocyte spheroid [18]. Biliatresone and decreases in GSH indirectly downregulated *Sox17* [17]. GSH levels were shown to drop in the first few hours but increased and recovered to normal levels at 24 h of biliatresone treatment. However, *Sox17* expression was still significantly lower in biliatresone-treated mouse cholangiocytes at 24 h of treatment [18]. Taken all these together, this suggests that a transient drop in GSH level is sufficient to induce a downregulation of *SOX17*, and a later recovery of GSH level is unable to restore the normal *SOX17* expression. Glutathione depletion has also been shown to induce premature cell senescence in retinal pigment epithelial cells [44]; future experiments are needed to address if biliatresone also induces senescence as one of the molecular mechanisms underlying its adverse effects in liver organoids.

In summary, biliatresone-induced cholangiocyte injury includes a decrease in cell–cell tight junctions, polarity changes, increased epithelial permeability and loss of cilia and cilia function in human cholangiocytes. The biliatresone-induced morphological and developmental changes may provide important insights into BA pathogenesis in humans and may lead to the development of new preventive measures for environmental toxins or novel treatments. Although biliatresone is unlikely to be relevant to human BA, given its limited distribution, it provides evidence that prenatal exposure to a toxin could lead to BA in neonates while sparing their mothers. There are also preliminary data suggesting that a toxin found in harmful algal blooms, microcystin-RR, is a biliary toxin. This toxin, like biliatresone, is specific to neonatal (as opposed to adult) cholangiocytes and causes redox stress in cholangiocytes [45]. Maternal exposure to aflatoxin B1 and its toxic metabolite aflatoxin B1-8,9 epoxide [46] has been suggested to cause BA [47,48]. Normally, the epoxide is detoxified via Glutathione S-Transferase Mu 1 (GSTM1)-mediated conjugation to GSH. In this study, a significant percentage of infants with BA were GSTM1 null and had detectable aflatoxin in liver cores [47,48]. Intriguingly, like biliatresone, aflatoxin and its epoxide metabolite are potent electrophiles, and their toxicity actions are also linked to GSH. Pregnant women may be exposed to other GSH-depleting toxins, and some of these may cross the placenta, causing oxidative stress and injury to the fetal hepatobiliary system, contributing to immune dysregulation in BA development. Human liver-derived organoids can serve as a human proxy for screening of these possible GSH-depleting toxins.

This study is not without limitations. The human liver tissue-derived liver organoids used in the current study mainly comprise cholangiocytes, which resemble the structural and physiological characteristics of cholangiocytes and have been used to study BA pathogenesis [27,29]. However, advanced multicellular liver organoids comprising hepatocytes, cholangiocytes and non-parenchymal cells (Kupffer cells, endothelial cells and stellate cells) are needed in the future to better mimic the complex multicellular nature of the liver to delineate the precise pathological roles of toxins in BA and to screen for hepatobiliary toxins.

In conclusion, the current study showed for the first time that the plant toxin biliatresone induces BA-like morphological changes in human cholangiocytes, suggesting that environmental toxins could contribute to BA pathogenesis in humans.

## 4. Materials and Methods

### 4.1. Human Ethics

Liver biopsies of children with non-BA liver diseases, including non-tumor liver of hepatoblastoma (HB; n = 3) and choledochal cyst (CC; n = 4), were obtained during operations between 2018 and 2023 with full informed consent from parents or guardians. This study was approved by the Institutional Review Board of the University of Hong Kong/Hospital Authority Hong Kong West Cluster (HKU/HA HKW IRB) (UW 16-052; Date of approval: 27 January 2016 and UW 20-757; Date of approval: 14 August 2020). All participants were interviewed, given written consents, and signed informed consents were obtained. The purpose of this research study was clearly explained, and all participants were allowed at least 10 min to ask questions, if any, and make decisions. Their treatment would not be altered regardless of their willingness to participate in this study. Patients’ information for organoids is shown in Appendix A.

### 4.2. Human Liver Organoids

Generation, passaging and expansion of liver organoids from liver biopsies were performed following our previously described protocol [27]. A total of 130 organoids were generated from the three HB patients, and 200 organoids were generated from the four CC patients for the current experiment. Briefly, liver biopsy was minced and digested in a gentleMACS-C Tube with 5 mL of digestion medium (Multi Tissue Dissociation Kit 1), filtered (70 µm and 30 µm) and sorted using human CD326 (EpCAM) magnetic beads. CD326-positive cells were mixed with Matrigel (50,000 cells in 50 µL) and added to each well of a four-well culture plate (Nunc™ 4-Well Dishes, Roskilde, Denmark). After Matrigel solidification, organoid medium (Advanced DMEM/F12 supplemented with 1% Penicillin/Streptomycin (Invitrogen, Waltham, MA, USA), 250 ng/mL Amphotericin B (GIBCO, Waltham, MA, USA), 25 µM HEPES, 1% N2 and 1% B27 (GIBCO), 1.25 mM N-Acetylcysteine (Sigma, St. Louis, MO, USA), and 10 nM gastrin (Sigma), and the growth factors are as follows: 50 ng/mL mEGF (Peprotech, Rocky Hill, NJ, USA), 100 ng/mL FGF10 (Peprotech), 25 ng/mL HGF (Peprotech), 10 mM Nicotinamide (Sigma), 5 µM A83.01 (Tocris, Bristol, UK), 10 µM FSK (Tocris), 25 ng/mL Noggin (Peprotech), 500 ng/mL R-Spondin 1 (R&D, Minneapolis, MN, USA), 100 ng/mL Wnt3a (R&D), and 10 µM Y27632 (Sigma Aldrich) were added for organoid culturing. The medium was changed every three days.

### 4.3. Treatment of Biliatresone

Biliatresone concentrations ranging from 0.125 µg/mL to 2 µg/mL were previously shown to induce different degrees of aberrant growth of mouse liver organoids, and the 2 µg/mL was the optimal concentration for causing lumen obstruction [13]. Therefore, in the current study to investigate the effects of biliatresone, after culturing for 5 days, biliatresone (final concentration: 2 µg/mL; 2867, Axon MEDCHEM; stock solution: 2 mg/mL in DMSO) was added to the medium, and the organoids were cultured for different periods for subsequent analysis. For the untreated control culture, DMSO (same volume as biliatresone) was added to the culture. 

### 4.4. Measurement of Organoid Growth

To establish the growth curve of control and biliatresone-treated organoids, pictures of all the organoids from both groups were taken under the same magnification on Days 0, 3 and 5 after treatment. The diameters of the organoids (25 randomly chosen organoids were measured in each culture) were then measured and calculated from the pictures using the measurement tool in Photoshop CS6. The results were expressed as Mean ± SD to indicate the size of the organoid. 

### 4.5. Immunostaining and Confocal Imaging

Organoids in the Matrigel were washed with PBS (phosphate-buffered saline) before being fixed with 4% paraformaldehyde at room temperature for 20 min. The organoids were permeabilized with Triton X-100 (0.5% in PBS) for 20 min and blocked with 3% bovine serum albumin (BSA, Sigma-Aldrich, Waltham, MA, USA) in PBS with 0.05% Triton X-100 (PBST) for 30 min at room temperature. The organoids were then incubated in primary antibodies (diluted in PBST with 3% BSA) at 4 °C overnight. After PBST washes (5 min for 3 times), organoids were incubated in secondary antibodies (diluted in PBST with 3% BSA) for 2 h at room temperature. Afterward, the organoids were rinsed with PBS 3 times and incubated with DAPI solution for 30 min at room temperature. The organoids were rinsed with PBS (5 min for 3 times). Primary and secondary antibodies were anti-ZO-1 (1:50, RA231621, Thermo Fisher), anti-CK19 (1:200, ab220193, Abcam, Cambridge, UK), anti-F-actin (1:500, A22287, Thermo Fisher), anti-HNF4A (1:50, ab201460, Abcam), Alexa Fluor 488 (1:500, A11008, Invitrogen) and Alexa Fluor 594 (1:500, A11032, Invitrogen).

### 4.6. Confocal Imaging and Analysis

Organoids were imaged on a laser confocal microscope (Leica, Wetzlar, Germany) equipped with a 20× dry objective and a 63× oil-immersion objective. Images were analyzed using ImageJ software. For quantification of the number of cilia from each group, the organoids were imaged using confocal microscopy (0.8 μm *Z*-axis interval, 40 μm in thickness). Maximum intensity projection was performed to visualize the cilia in the organoids. 

### 4.7. Rhodamine 123 Transport Assay

Liver organoids were released from Matrigel using cold Advanced DMEM/F12 and resuspended in culture medium with or without Verapamil (10 μM) and incubated at 37 °C for 30 min. Afterward, the liver organoids were washed with PBS and incubated in a culture medium containing Rhodamine 123 (100 μM) at 37 °C for 30 min. After washing with PBS, the fluorescence of liver organoids was visualized using laser confocal microscopy (Leica) immediately (at 0 min) and after 30 min of incubation in the culture medium.

### 4.8. FITC-Labeled Dextran Diffusion Tests

For testing the integrity of liver organoids, the liver organoids were released from Matrigel using cold Advanced DMEM/F12 and resuspended in a culture medium containing FITC-labeled dextran of 10 kDa (10 μM). The fluorescent intensity of liver organoids was imaged using laser confocal microscopy at 10 and 30 min.

### 4.9. Calcium Imaging

For calcium imaging, the organoids were seeded onto Matrigel-coated 6-channel ibidi microfluidic chips (80,607 ibidi; around 5000 cells were seeded into each channel), and the microfluidic chips were incubated at 37 °C in an incubator for 3 days. Biliatresone treatment was started by adding biliatresone (final concentration: 2 µg/mL; 2867, Axon MEDCHEM; stock solution: 2 mg/mL in DMSO) into the medium on Day 3, and the chips were cultured for 5 days with daily medium changes. For the untreated control culture, DMSO (same volume as biliatresone) was added to the culture. To prepare for the calcium influx measurement, cholangiocyte monolayers were incubated with 5 μM Calbryte 520-AM (ATT) (prepared in Hanks’ balanced salt solution, HBSS) for 1 h at 37 °C. Before the flow experiment, the cholangiocytes were washed with HBSS (5 min for 3 times). The 6-channel chip was connected to a syringe, which was mounted on a syringe pump to apply shear flow with HBSS at 1 dyne/cm^2^. Time-lapse images were acquired for green fluorescence for 20 s (1-s intervals and 10 milliseconds exposure time) using a Nikon Ti2e microscope (Nikon, Tokyo, Japan). 

### 4.10. Statistical Analysis 

For each experiment using organoids, organoids generated from at least two different patients were used as biological replicates, and the experiment using organoids from each patient was performed in triplicate. All data were shown as mean ± standard deviation (Mean ± SD). Data from separate quantitative analyses were compiled and analyzed between groups. The Shapiro–Wilk test was used to check for normality prior to performing the comparison using Student’s 2-tailed *t*-test, and *p* < 0.05 was regarded as statistically significant.

## Figures and Tables

**Figure 1 toxins-16-00144-f001:**
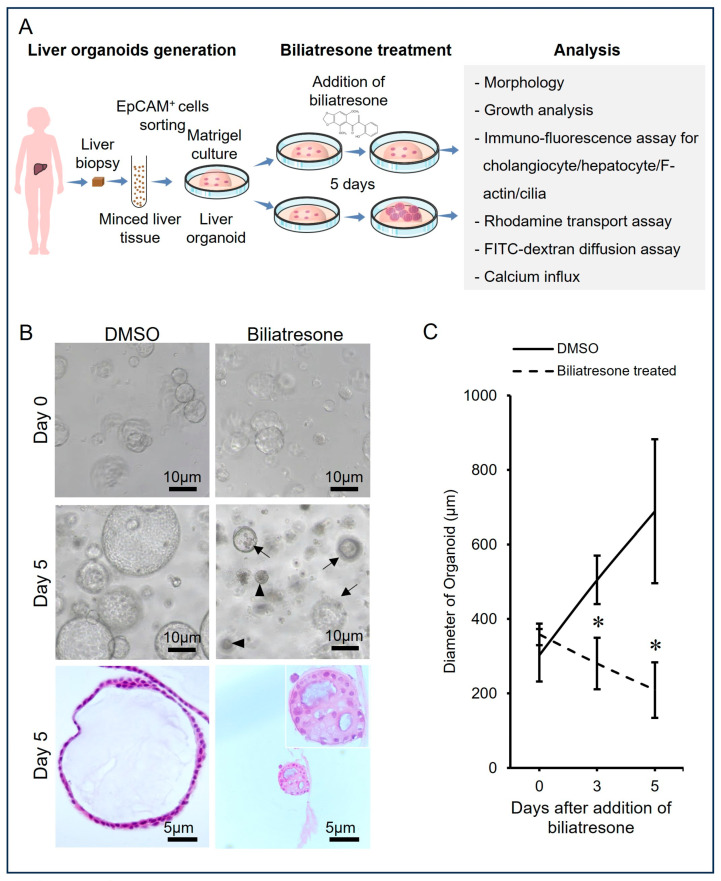
Biliatresone-induced aberrant growth of human liver organoids. (**A**) Schematic diagram of the establishment of human liver organoids for biliatresone treatment. (**B**) Representative bright-field images; Hematoxylin & Eosin staining of sections of control and Day-5 biliatresone-treated organoids. (**C**) Growth curve of control and biliatresone-treated human liver organoids. The diameters of organoids (25 randomly chosen organoids were measured in each culture) were measured for the growth curve. *, a statistically significant difference between biliatresone-treated and untreated cultures (*p* < 0.05). Error bars indicate standard deviation.

**Figure 2 toxins-16-00144-f002:**
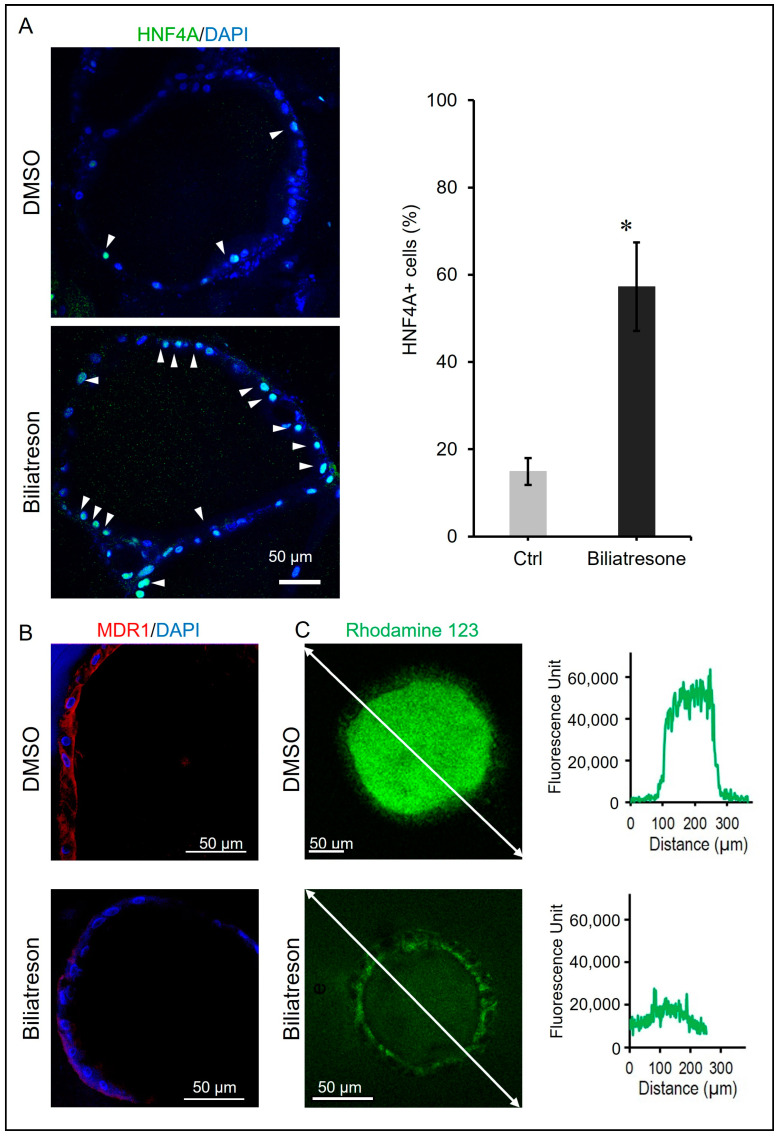
Biliatresone-induced hepatocytic differentiation of human liver organoids. (**A**) Representative images of control and Day-2 biliatresone-treated human liver organoids stained for HNF4A (green). Nuclei were stained with DAPI (blue). The percentage of HNF4A-positive cells (white arrow) in control and Day-2 biliatresone-treated organoids was determined by counting the total number of cells and HNF4A+ cells of the organoids. The number of organoids counted in each group = 3; *, *p* < 0.05, Student’s *t*-test; error bars indicated the standard deviation. (**B**) Representative images of control and Day-2 biliatresone-treated human liver organoids stained for MDR1 (red). Nuclei were stained with DAPI (blue). (**C**) MDR1 activity assay of organoids. (**C**) Representative images demonstrating the fluorescent substrate Rhodamine 123 (R123) localization in the lumen of control organoids at 30 min. In contrast, luminal R123 localization of Day-2 biliatresone-treated organoids was minimal during the 30-min incubation. The fluorescent intensity along the white line of images of control and Day-2 biliatresone-treated organoids at 30 min of incubation were plotted.

**Figure 3 toxins-16-00144-f003:**
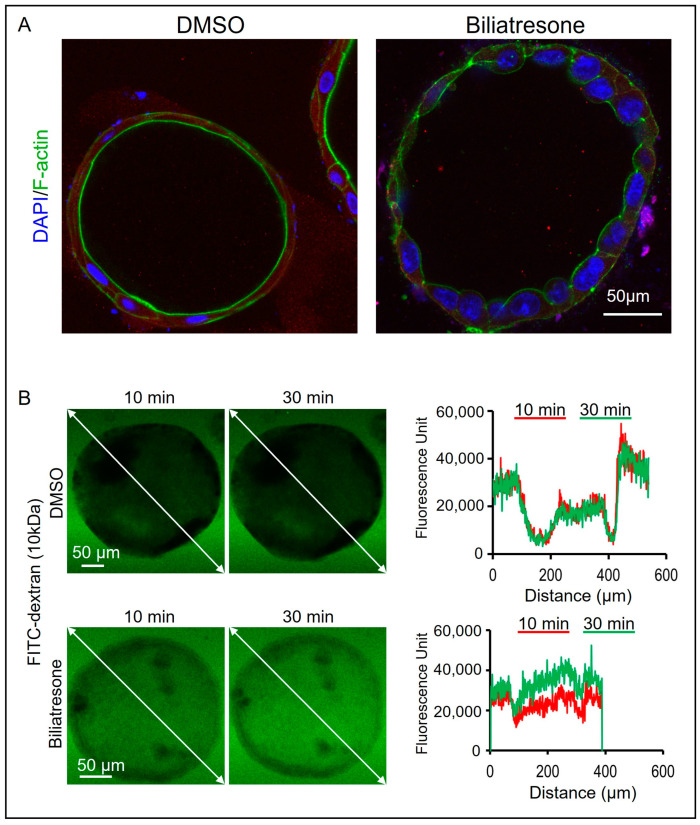
Biliatresone-induced cytoskeleton defects in human liver organoids. (**A**) Immunostaining of control and Day-2 biliatresone-treated human liver organoids for cytoskeleton protein F-actin (green). Nuclei were stained with DAPI (blue). The number of organoids examined in each group was 20. (**B**) (**Left panel**). Representative images demonstrating the accumulation of FITC-dextran localization in the lumen of Day-2 biliatresone-treated human liver organoids but not of control organoids at 10 and 30 min of incubation. (**Right panel**), the fluorescent intensity along the white line of images of control and Day-2 biliatresone-treated organoids at 10 and 30 min of incubation were plotted. The number of organoids examined in each group was 10.

**Figure 4 toxins-16-00144-f004:**
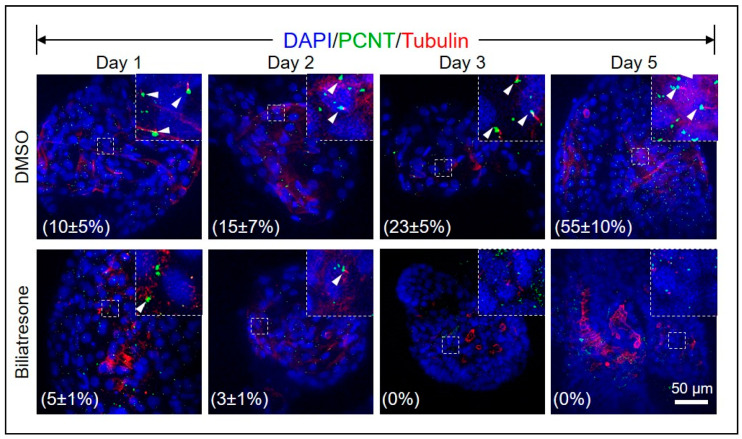
Biliatresone decreased the number of ciliated cholangiocytes in human organoids. Representative images of control and Day-1, -2, -3 and -5 biliatresone-treated human liver organoids stained for pericentrin (PCNT; green) and acetylated α-tubulin (Tubulin; red). Nuclei were stained with DAPI (blue). Boxed regions were magnified and shown as insets. The percentages of ciliated cells (white arrow) in control and biliatresone-treated organoids were determined by counting the total number of cells and ciliated cells of the organoids. The number of organoids counted in each group was 20.

**Figure 5 toxins-16-00144-f005:**
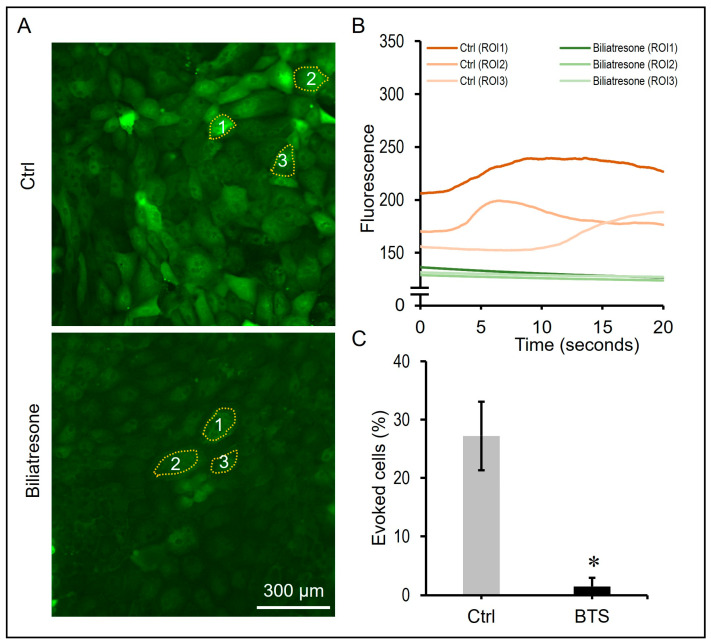
Reduction in cholangiocyte cilia mechanosensory response in biliatresone-treated organoids. (**A**) Representative images of control and biliatresone-treated human cholangiocytes stained with Calbryte 520 after application of shear flow with HBSS at 1 dyne/cm^2^. (**B**) Plot profile of fluorescent intensity changes of selected control and biliatresone-treated human cholangiocytes (highlighted with dotted line and numbered 1, 2 and 3) for 20 s. (**C**) The percentages of evoked cells after perfusion stimulation of control (Ctrl) and biliatresone-treated (BTS) human cholangiocytes were determined by counting the total number of cells and stimulated cells (green) at three randomly chosen fields of the cultures. *, *p* < 0.05, Student’s *t*-test; error bars indicated the standard deviation.

## Data Availability

All data generated or analyzed during this study are included in this published article (and its Appendix A).

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
