# Peer review of "Environmental Toxin Biliatresone-Induced Biliary Atresia-like Abnormal Cilia and Bile Duct Cell Development of Human Liver Organoids"

_toxins, 2024, doi:10.3390/toxins16030144_

Round 1

Reviewer 1 Report

Comments and Suggestions for Authors

This paper reports on the effects of biliatresone, an isoflavinoid that produces biliary atresia in Australian livestock and zebrafish. mouse cholangiocyte spheroids and ex-vivo bile duct cultures. In this study the authors use human liver oreganoids and expose then to a single dose of biiatresone which produces morphological changes somewhat similar to prior studies including  retarded growth, disrupted apical polarity ,loss of cilia and differentiation towards a hepatocyte linage  by seeing increasing expression of HNF4 A expression.

Comments;  1) It is difficult to know if the findings in this study are specific for changes seen in other models of biliary atresia with this compound dor whether these are "toxic nonspecific changes". 

2) Only one dose of biliatresone is used in this study and it would have been useful to perform a dose related study to see if there is a gradient of changes with increasing doses.

3) Some  of the images (fig 1 B day 5) suggest that the compound may be inducing senescence and markers of senesence should be performed to assess this possibility.

4) It is noteworthy that glutathione was reduced only by a small amount and increased to normal levels with time in culture. Depletions of glutathione are thought to play a role in the toxicity of biliatresone  in the zeberafish and the authors should discuss this discrepancy in their model.

5) Reference 15 should be broken down to specfic references for  the various models described.

Author Response

Authors thanks for the reviewers’ insightful comments and are delighted to be given a chance to revise and resubmit our manuscript. Our point-to-point responses to the reviewers’ comments are detailed as shown below:

Reviewer 1

Comment: It is difficult to know if the findings in this study are specific for changes seen in other models of biliary atresia with this compound or whether these are "toxic nonspecific changes". 

Response: Our current finding together with other publications on biliatresone and other biliary toxins seem to suggest that depletion of GSH and subsequent oxidative stress is a common mechanism underlying the pathological role of these toxins in BA. A new paragraph is added to the Discussion to highlight this (P.18 ln18 to P.19 ln10).

Comment: Only one dose of biliatresone is used in this study and it would have been useful to perform a dose related study to see if there is a gradient of changes with increasing doses.

Response: Biliatresone concentrations ranging from 0.125 µg/ml to 2 µg/ml were previously shown to induce different degree of aberrant growth of mouse liver organoid, and the 2 µg/ml was the optimal concentration in causing lumen obstruction [13]. Therefore, in the current study, we chose the use the optimal concentration of biliatresone (2 mg/ml) to investigate the effects of biliatresone on human liver organoids (P.7 ln11 to ln17).

Comment: Some of the images (fig 1B day 5) suggest that the compound may be inducing senescence and markers of senescence should be performed to assess this possibility.

Response: Thanks for the suggestion. However, due to the scarcity of HB and CC cases in our hospital, we have not been able to recruit new patients to generate new organoids to investigate if senescence is induced in biliatresone-treated organoids in a short time. We have previously performed β-galactosidase staining on BA liver organoids (BA liver organoids also displayed poor growth, unexpanded features with a cholangiocyte to hepatocytic shift of differentiation same as biliatresone-treated organoids[1]) and showed there is no β-galactosidase staining on BA liver organoids, suggesting that aberrant organoid development is not caused by senescence. Since biliatresone-treated organoids displayed similar aberrant features as BA organoids, we believe that biliatresone unlikely induces senescence in organoids. A sentence regarding the possibility of senescence in biliatresone-treated organoids was added to the Discussion of the revised manuscript (P.18 ln9 to ln12).

Comment: It is noteworthy that glutathione was reduced only by a small amount and increased to normal levels with time in culture. Depletions of glutathione are thought to play a role in the toxicity of biliatresone in the zebrafish and the authors should discuss this discrepancy in their model.

Response: Authors apologize not having explained clearly in the manuscript that biliatresone has also been showed to cause a rapid and transient decrease in GSH, which was both necessary and sufficient to mediate its reduction of Sox17 expression and its effects in mouse cholangiocyte spheroid[2]. A new sentence is added to the Discussion to highlight this (P.17 ln21 to P.18 ln3).

Comment: Reference 15 should be broken down to specific references for the various models described.

Response: Done as suggested.

References

  1. Babu, R.O.; Lui, V.C.H.; Chen, Y.; Yiu, R.S.W.; Ye, Y.; Niu, B.; Wu, Z.; Zhang, R.; Yu, M.O.N.; Chung, P.H.Y.; et al. Beta-amyloid deposition around hepatic bile ducts is a novel pathobiological and diagnostic feature of biliary atresia. J Hepatol 2020, 73, 1391-1403, doi:10.1016/j.jhep.2020.06.012.
  2. Waisbourd-Zinman, O.; Koh, H.; Tsai, S.; Lavrut, P.M.; Dang, C.; Zhao, X.; Pack, M.; Cave, J.; Hawes, M.; Koo, K.A.; et al. The toxin biliatresone causes mouse extrahepatic cholangiocyte damage and fibrosis through decreased glutathione and SOX17. Hepatology 2016, 64, 880-893, doi:10.1002/hep.28599.

Reviewer 2 Report

Comments and Suggestions for Authors

- Introduction (line 39-41): please, double check the syntax of the sentence. 

- In the final paragraph of the introduction, the authors should just describe the general objective. They should not summarize the main results, which should actually represent the incipit of the discussion.

- Conversely, some more biochemical, mechanistic, and biological information on biliatresone would be useful. Moreover, it is not clear if it can be found in specific products used in agriculture or industry and, then, expose people on a large scale.

- Methods are described well overall, but there are some important points that are unlear or lacking, as explained below. However, a figure graphically summarizing the process would add clarity and value to this manuscript. Probably, Figure 1a may be used, at least as a part of this revised figure.

- Another very important point in the methods is to create a specific ethical statement subsection. First of all, IRB approval date, should be disclosed. The study period must be also declared. The detailed and precise procedure of informed consent should be described. 

- The authors may also clarify why only children were chosen to create organoids from liver biopsies. 

- It is not clear how many organoids were created and/or how many patients were recruited. 

- Following the previous point, it is not clear how many experiment replications were done and, accordingly, how these were statistically analyzed (since there is no description of statistical methods). 

- Indeed, the results are very narrative and there are no values reported in general. 

- The discussion has not been reviewed from my side, yet, since methodological and result clarifications are needed. 

Comments on the Quality of English Language

see above

Author Response

Authors thanks for the reviewers’ insightful comments and are delighted to be given a chance to revise and resubmit our manuscript. Our point-to-point responses to the reviewers’ comments are detailed as shown below:

Reviewer 2

Comment: Introduction (line 39-41): please, double check the syntax of the sentence. 

Response: Revised as suggested.

Comment: In the final paragraph of the introduction, the authors should just describe the general objective. They should not summarize the main results, which should actually represent the incipit of the discussion.

Response: Revised as suggested.

Comment: Conversely, some more biochemical, mechanistic, and biological information on biliatresone would be useful. Moreover, it is not clear if it can be found in specific products used in agriculture or industry and, then, expose people on a large scale.

Response: A paragraph has been added to the Introduction and Discussion to include the required additional information of biliatresone in the revised manuscript (P.4 ln21 to P.5 ln3; P.18 ln18 to P.19 ln10).

Response: Methods are described well overall, but there are some important points that are unclear or lacking, as explained below. However, a figure graphically summarizing the process would add clarity and value to this manuscript. Probably, Figure 1a may be used, at least as a part of this revised figure.

Response: A revised Figure 1 is included to summarize the study as suggested.

Comment: Another very important point in the methods is to create a specific ethical statement subsection. First of all, IRB approval date, should be disclosed. The study period must be also declared. The detailed and precise procedure of informed consent should be described. 

Response: Information is added to the revised Materials and Methods section (P.6 ln2 to ln14). Patient information is included as Supplementary Table 1 in the revised manuscript.

Comment: The authors may also clarify why only children were chosen to create organoids from liver biopsies. 

Response: BA is a rare disease of the liver and bile ducts that occurs in infants, we generated liver organoids from liver biopsies of age-matched non-BA infants to (i) address if biliatresone has an adverse effect on the differentiation and functions of human cholangiocytes; and (ii) provide evidence for an etiological/pathobiological role of biliatresone exposure in BA in humans (P.5 ln 5 to ln19).

Comment: It is not clear how many organoids were created and/or how many patients were recruited. 

Response: Number of patients recruited and number of organoids created are indicated in the revised manuscript (P.6 ln16 to ln19). Number of organoids used in each experiment was added to the Figure legend.

Comment: Following the previous point, it is not clear how many experiment replications were done and, accordingly, how these were statistically analyzed (since there is no description of statistical methods).

Response: Number of experiment and statistical analysis are added in the revised Materials and Methods (P.10 ln10 to ln14).

Comment: Indeed, the results are very narrative and there are no values reported in general. 

Response: Values are added to the results of the revised manuscript.

Comment: The discussion has not been reviewed from my side, yet, since methodological and result clarifications are needed.

Response: The entire manuscript has been intensively revised as suggested by both reviewers to address all the comments.

Round 2

Reviewer 1 Report

Comments and Suggestions for Authors

Thank you for your corerctions and additions

Author Response

Authors thank for the comments and suggestions for the revision, which has improved the manuscript tremendously.

Reviewer 2 Report

Comments and Suggestions for Authors

-“All participants were interviewed and informed consents were obtained”. Study participants are children, then I guess the authors refer to the guardians, which should have given written and signed informed consent. This information should be clear in the manuscript.

- The statistical analysis description is not complete. Why did the authors assume a normal distribution, in order to use a parametric test? Was the normality test with any specific test? In case of appropriate use of t-test, was it 2-tailed?

- Can you also better explain the strategy (“Organoids generated from at least two different patients were used for biliatresone treatment in two separate experiments. Each experiment was performed in triplicate well”), in order to have a clear information about the number of different observations (patients/organoids) for each experiment?

- Why did you use “Mean/S.E.M.” as descriptive statistics? By the way, these aspects of descriptive statistics should be also mentioned in the methods. Indeed, elsewhere mean/SD was used. Why did you use different descriptive strategy?

- In figure 1c, the meaning of the asterisk should be described.

- Following the comments of the previous round of review, I do not see many numerical and statistical values added in the results description. This major change should involve all results subsections.

- As regards the discussion, the authors should list the main novelty/findings at the beginning, which then should be discussed individually in light of the current literature.

- A separate conclusion section should be created. It should briefly highlight the main novelty and conclusion supported by the discussion of the current results.

- I do not see any discussion on the study limitations, which may limit its translational significance.

Comments on the Quality of English Language

see above

Author Response

Comment: All participants were interviewed and informed consents were obtained”. Study participants are children, then I guess the authors refer to the guardians, which should have given written and signed informed consent. This information should be clear in the manuscript. 

Response: This information has been added to the revised version (ln 68 & ln 72).

Comment: The statistical analysis description is not complete. Why did the authors assume a normal distribution, in order to use a parametric test? Was the normality test with any specific test? In case of appropriate use of t-test, was it 2-tailed?

Response: Because the distribution of data is close to mean, that is the reason why we assumed a normal distribution. Shapiro-Wilk test was used to check for normality prior to performing the comparison between groups. We used Student’s 2-tailed t test and p<0.05 was regarded as statistical significance. This information has been added to the revised Materials & Methods (ln 161 to ln 163).

Comment: Can you also better explain the strategy (“Organoids generated from at least two different patients were used for biliatresone treatment in two separate experiments. Each experiment was performed in triplicate well”), in order to have a clear information about the number of different observations (patients/organoids) for each experiment?

Response: For each experiment using organoids, organoids generated from at least two different patients were used as biological replicates, and experiment using organoids from each patient was performed in triplicate. This clarification has been added to the revised Materials & Methods (ln 158 to ln 160).

Comment: Why did you use “Mean/S.E.M.” as descriptive statistics? By the way, these aspects of descriptive statistics should be also mentioned in the methods. Indeed, elsewhere mean/SD was used. Why did you use different descriptive strategy? 

Response: Authors apologize the inconsistency. To be consistent, we redo the Figure 1C in the revised version using mean±SD which was the only figure using mean±S.E.M. Information of descriptive statistics has also been added to the revised Materials & Methods (ln 160 to ln 161).

Comment: In figure 1c, the meaning of the asterisk should be described. 

Response: * indicates statistical significant difference between biliatresone-treated and untreated cultures (p<0.05) (ln 188 to ln 189).

Comment: Following the comments of the previous round of review, I do not see many numerical and statistical values added in the results description. This major change should involve all results subsections. 

Response: Numerical and statistical values have been added to each result subsections in the revised manuscript (highlighted in grey).

Comment: As regards the discussion, the authors should list the main novelty/findings at the beginning, which then should be discussed individually in light of the current literature.

Response: Main novelty/findings of the current study has been added to the beginning of the Discussion of the revised manuscript (ln 320 to ln 322; ln 325 to ln 328).

Comment: A separate conclusion section should be created. It should briefly highlight the main novelty and conclusion supported by the discussion of the current results. 

Response: Added as suggested (ln 420 to ln 422).

Comment: I do not see any discussion on the study limitations, which may limit its translational significance.

Response: A paragraph on the limitations of the current have been added to the revised manuscript (ln 412 to ln 419).